# CogniMap3D: Cognitive 3D Mapping and Rapid Retrieval

**Feiran Wang**[1]**, Junyi Wu**[1]**, Dawen Cai**[2]**, Yuan Hong**[3]**, Yan Yan**[1][†]
[1]University of Illinois Chicago, [2]University of Michigan, [3]University of Connecticut

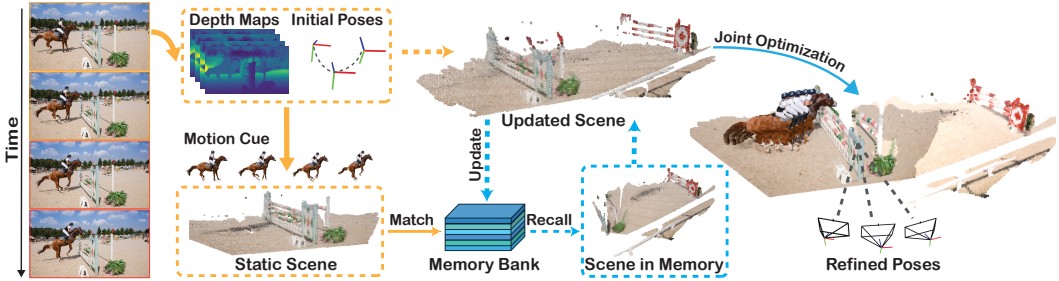

Figure 1: CogniMap3D maintains a cognitive mapping system that recalls, stores, and updates memories. Given an input video, it outputs camera poses and point clouds by isolating static scenes through motion cues, interacting with its memory bank, and optimizing across multiple visits.

## Abstract

We present CogniMap3D, a bioinspired framework for dynamic 3D scene understanding and reconstruction that emulates human cognitive processes. Our approach maintains a persistent memory bank of static scenes, enabling efficient spatial knowledge storage and rapid retrieval. CogniMap3D integrates three core capabilities: a multi-stage motion cue framework for identifying dynamic objects, a cognitive mapping system for storing, recalling, and updating static scenes across multiple visits, and a factor graph optimization strategy for refining camera poses. Given an image stream, our model identifies dynamic regions through motion cues with depth and camera pose priors, then matches static elements against its memory bank. When revisiting familiar locations, CogniMap3D retrieves stored scenes, relocates cameras, and updates memory with new observations. Evaluations on video depth estimation, camera pose reconstruction, and 3D mapping tasks demonstrate its state-of-the-art performance, while effectively supporting continuous scene understanding across extended sequences and multiple visits.

## 1 Introduction

Humans exhibit a remarkable ability to process dynamic visual scenes: our attention naturally prioritizes moving objects while simultaneously constructing persistent spatial representations of static environments (Abrams & Christ, 2003; Franconeri & Simons, 2003). For instance, during an equestrian performance shown in Fig. 1, observers unconsciously notice the motion of horse and rider, while extracting depth cues and motion parallax to separate moving objects from static backgrounds (Rogers & Graham, 1979; Born & Bradley, 2005). Based on spatial representations, the hippocampus constructs internal "cognitive maps" in an egocentric reference frame (Eichenbaum, 2015; Burgess, 2006). When revisiting familiar environments, humans reliably recall static scenes even when dynamic elements have changed, facilitating efficient navigation and spatial reasoning with minimal cognitive load (O'keefe & Nadel, 1979; Epstein et al., 2017).

---

[†]Corresponding Author
[*]The code is available at `https://github.com/Brack-Wang/cognimap3D`.

Inspired by these human cognitive processes, we aim to build 3D cognitive mapping systems that can similarly distinguish dynamic objects from static backgrounds while maintaining persistent memory of static 3D environments. The challenge lies in developing systems that can simultaneously: (1) distinguish between static and dynamic scene elements in monocular videos, (2) construct and maintain persistent representations of static environments and efficiently recall and update these representations when revisiting familiar scenes, and (3) establish stable camera pose estimates that remain geometrically consistent despite the presence of dynamic objects.

Recent advances in 3D reconstruction have made significant progress in related areas. Monocular depth estimation (MDE) works (Bian et al., 2021; Ranftl et al., 2021; Yin et al., 2022; Bhat et al., 2023; Godard et al., 2019; Yang et al., 2024; Li & Snavely, 2018) estimate precise 3D information but fail to localize camera poses. Visual SLAM approaches (Agarwal et al., 2011; Campos et al., 2021; Schonberger & Frahm, 2016; Davison et al., 2007; Pollefeys et al., 2008; Mur-Artal et al., 2015) achieve accurate camera poses but typically require additional camera intrinsics and precise initialization. Visual foundation models (VFM) (Wang et al., 2024; Zhang et al., 2024; Duisterhof et al., 2024; Wang et al., 2025b;a) directly regress 3D geometry and camera poses from RGB images, establishing a solid foundation for dynamic scene reconstruction. However, these approaches lack the cognitive mapping capabilities needed for persistent memory and scene revisitation.

Building on human cognitive mechanisms and recent advances in visual foundation models, we present CogniMap3D: Cognitive 3D Mapping and Rapid Retrieval, a comprehensive framework for dynamic scene understanding that emulates human cognitive processes with three key capabilities: 1) A multi-stage motion cue framework that accurately separates dynamic objects from static backgrounds through progressive refinement of 2D-3D motion cues; 2) A cognitive mapping system that creates, recalls, and updates memory of static environments, enabling efficient scene recognition and relocalization across multiple visits; 3) A geometrically consistent camera pose optimization strategy that stabilizes predicted parameters through factor graph optimization focused on static regions.

Specifically, given an image stream, CogniMap3D first predicts initial camera parameters and depth information through a Visual Foundation Model (Wang et al., 2025a). Our motion cue framework then progressively identifies dynamic objects through a coarse-to-fine approach combining optical flow clustering, geometry-based motion analysis, and 3D keypoint refinement. With the accurate static scene representation, our system efficiently matches hybrid features from 2D keyframes and 3D static geometry against the memory bank, verifying potential matches through geometric alignment of static point clouds. Upon recognizing a familiar environment, CogniMap3D recalls the stored static scene, relocates the camera pose, and updates the memory with new observations, enabling continuous scene refinement. To enhance geometric consistency, we implement a factor graph optimization that jointly refines camera poses using constraints from both newly observed static regions and updated memory.

We evaluate CogniMap3D on various 3D tasks, including consistent video depth estimation, camera pose reconstruction, and 3D reconstruction, achieving competitive or state-of-the-art performance. Our experiments demonstrate the system's ability to efficiently recall previously stored environments, update them with new observations, and maintain a coherent memory bank that supports continuous scene understanding across extended sequences and multiple visits to the same scene.

## 2 RELATED WORK

**Foundation Models for 3D Reconstruction.** Directly predicting 3D geometry from RGB images offers significant flexibility for real-world applications. Monocular depth estimation (MDE) (Bian et al., 2021; Ranftl et al., 2021; Yin et al., 2022; Bhat et al., 2023; Godard et al., 2019; Yang et al., 2024; Li & Snavely, 2018) has demonstrated robust generalization across diverse scenes but lacks camera pose information and temporal consistency in videos. DUSt3R (Wang et al., 2024) pioneered a pointmap representation for scene-level 3D reconstruction, implicitly inferring both camera pose and aligned point clouds from image pairs. Subsequent approaches (Lu et al., 2024; Zhang et al., 2024; Sucar et al., 2025; Wang & Agapito, 2024; Duisterhof et al., 2024) extended this framework but required processing videos as numerous image pairs with time-consuming optimization. Recent advances toward online processing include CUT3R (Wang et al., 2025b), which implements a stateful recurrent network for incremental pointmap refinement, and VGGT (Wang et al., 2025a), which employs a feed-forward model for joint prediction of camera poses and 3D geometry without

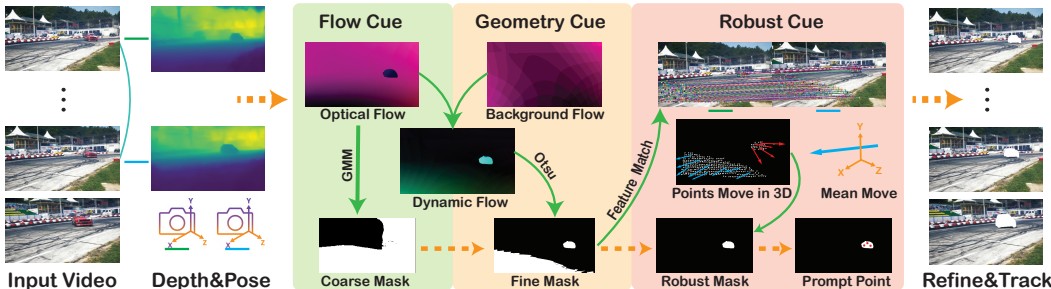

Figure 2: **Multi-stage Motion Cue for Locating Dynamic Area.** Given a pair of images in video, we first predict the initial depth and camera pose through VFM to establish 3D prior. Our pipeline then processes three specialized motion cues through progressive 2D-3D interaction effectively isolates robust dynamic regions, enabling accurate refinement and tracking across subsequent frames.

post-processing. However, these visual foundation models focus mainly on immediate observations without mechanisms for persistent scene understanding.

**Dynamic Scene Reconstruction.** Visual foundation models (VFM) for dynamic scene reconstruction (Zhang et al., 2024; Team et al., 2025; Ravi et al., 2024; Chen et al., 2025) aim to recover 3D geometry when both camera and scene elements are in motion. Recent approaches exhibit distinct trade-offs: MonST3R (Zhang et al., 2024) extends DUSt3R through optical flow but employs threshold-based detection with limited generalization; MegaSAM (Li et al., 2024) utilizes neural networks for motion prediction but suffers from domain transfer issues; AETHER (Team et al., 2025) leverages SAM2 (Ravi et al., 2024) for segmentation but struggles with elements out of preset category; and BA-Track (Chen et al., 2025) implements 3D tracking-based decoupling but requires camera intrinsic priors. Our multi-stage motion cue framework achieves robust dynamic-static separation through progressive refinement, combining optical flow clustering, geometry-based motion analysis, and 3D keypoint refinement.

**Structure from Motion and Visual SLAM.** Classical SLAM methods (Agarwal et al., 2011; Campos et al., 2021; Schonberger & Frahm, 2016; Davison et al., 2007; Pollefeys et al., 2008; Mur-Artal et al., 2015) estimate camera poses through feature matching and bundle adjustment but struggle with textureless regions. Learning-based approaches like Droid-SLAM (Teed & Deng, 2021) advance differentiable optimization yet exhibit limited generalization to dynamic environments. Recent developments for handling dynamic scenes include MegaSAM's (Li et al., 2024) probability predictions, DPVO's (Teed et al., 2023) patch-based features, MASt3R-SfM's (Duisterhof et al., 2024) learned feature integration, Anycam's (Wimbauer et al., 2025) depth-optical flow combination, and BA-Track's (Chen et al., 2025) point decoupling. However, these systems typically require camera intrinsics and maintain internal states that conflict with VFM's predictions. Our factor graph optimization framework refines VFM-predicted camera poses using multi-view constraints on static regions, yielding consistent trajectories compatible with foundation model outputs.

## 3 METHOD

Our approach takes monocular videos as input to achieve dynamic scene understanding with persistent spatial memory. The pipeline consists of three integrated components: a multi-stage motion cue framework that identifies dynamic objects, a cognitive mapping system that creates, recalls, and retrieves memories, and a factor graph optimization strategy that refines camera trajectories.

### 3.1 MULTI-STAGE MOTION CUE FRAMEWORK

We propose an efficient pipeline for identifying dynamic objects across scenes in monocular videos with moving cameras, as illustrated in Figure 2. Given initial depth maps $D$, camera poses $E$ estimated by VGGT (Wang et al., 2025a), we implement a progressive refinement process.

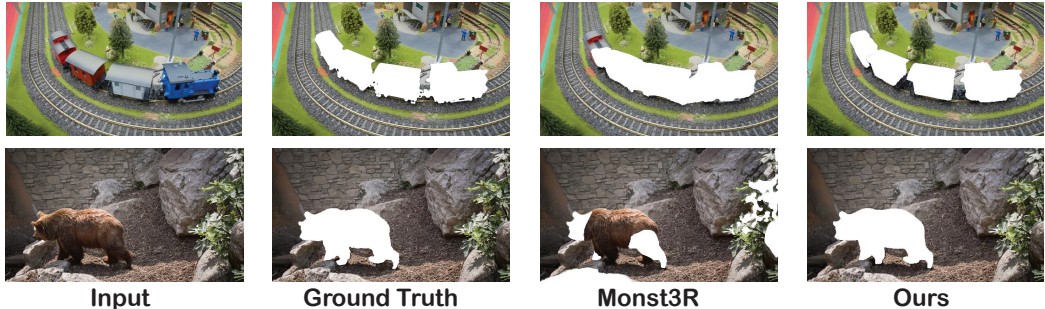

| Input | Ground Truth | Monst3R | Ours |

Figure 3: **Dynamic Mask Comparison.** We visualize dynamic regions as white overlays on input images. Compared with MonST3R, our method achieves more complete and precise masks.

**Flow Motion Cue.** To identify potential dynamic regions, we first compute the optical flow field $\mathbf{f}^{t\leftarrow t'} = \mathbf{F}_{\text{flow}}^{t\leftarrow t'}(I^t, I^{t'})$ and partition it into $K$ distinct components via Gaussian Mixture Model clustering. We then get coarse mask by excluding the component with minimal motion magnitude:

$$\mathcal{M}_{\text{flow}}^{t\leftarrow t'}(x,y) = \mathbb{1}\big(\ell\big(\mathbf{f}^{t\leftarrow t'}(x,y)\big) \neq \underset{i\in\{1,\dots,K\}}{\arg\min} \frac{1}{|\mathcal{C}_i|} \sum_{(u,v)\in\mathcal{C}_i} \big\|\mathbf{f}^{t\leftarrow t'}(u,v)\big\|\big), \qquad (1)$$

where $\ell(\cdot)$ assigns cluster labels and $\mathcal{C}_i = \{(u,v) \mid \ell(\mathbf{f}^{t\leftarrow t'}(u,v)) = i\}$ represents pixels in cluster.

**Geometry Motion Cue.** To distinguish moving objects from optical flow caused by camera movement, we follow a similar principle as MonST3R (Zhang et al., 2024) but with a more robust implementation. We unproject images into 3D pointmaps $P^t$ and $P^{t'}$, where $P(x,y) = D(x,y)(K^t)^{-1}[x,y,1]^\top$. By transforming $P^t$ through the relative camera pose and projecting onto $I^{t'}$'s image plane, we compute the expected flow for static scene elements. Subtracting this static scene flow prediction from the observed optical flow reveals motion caused by dynamic objects:

$$\mathbf{F}_{\text{res}}^{t\leftarrow t'}(x,y) = \left\| \mathbf{F}_{\text{flow}}^{t\leftarrow t'}(x,y) - \left[ \pi\left( K^{t'} E^{t'} (E^t)^{-1} P^t(x,y) \right) - (x,y) \right] \right\|, \qquad (2)$$

where $\pi(\cdot)$ denotes projection onto the image plane, $K^t$ is the intrinsic calibration matrix, $E^t$ the camera extrinsic parameters, and $P^t(x,y)$ the back-projected 3D point through corresponding depth $D(x,y)$. We derive the geometry motion cue $\mathcal{M}_{\text{geo}}^{t\leftarrow t'}(x,y) = \mathbb{1}(\mathbf{F}_{\text{res}}^{t\leftarrow t'}(x,y) > \tau)$ using Otsu's method to determine threshold $\tau$ automatically.

**Robust Motion Cue.** Leveraging dynamic region candidates from previous stages ($\mathcal{M}_{\text{flow}}^{t\leftarrow t'}$ and $\mathcal{M}_{\text{geo}}^{t\leftarrow t'}$), we further refine dynamic object localization by matching keypoints between frames and analyzing their correspondences. After transforming matched keypoints into world coordinates, we compute mean 3D displacement vector. Within candidate regions, keypoints whose displacement significantly deviates from this mean in either magnitude or direction are classified as dynamic. The final dynamic mask ($\mathcal{M}_{\text{dyn}}^{t\leftarrow t'}$) corresponds to regions containing these outlier keypoints.

**Dynamic Areas Tracking.** After precisely identifying dynamic areas, we uniformly extract prompt points within these regions, using SAM2 (Ravi et al., 2024) to refine dynamic masks and tracking dynamic areas across subsequent frames. Throughout the video, we continuously monitor the geometry motion cue $\mathcal{M}_{\text{geo}}$ for new moving objects, executing our pipeline when changes are detected and updating $\mathcal{M}_{\text{dyn}}$ accordingly. As shown in Fig. 3, our method provides accurate dynamic segmentation that serves as a foundation for both cognitive mapping and camera pose optimization.

## 3.2 COGNITIVE MAPPING SYSTEM

Inspired by the human ability to retain and recall static elements of familiar scenes, we design a cognitive map system that stores, recalls, and updates 3D scene representations, shown in Fig. 4.

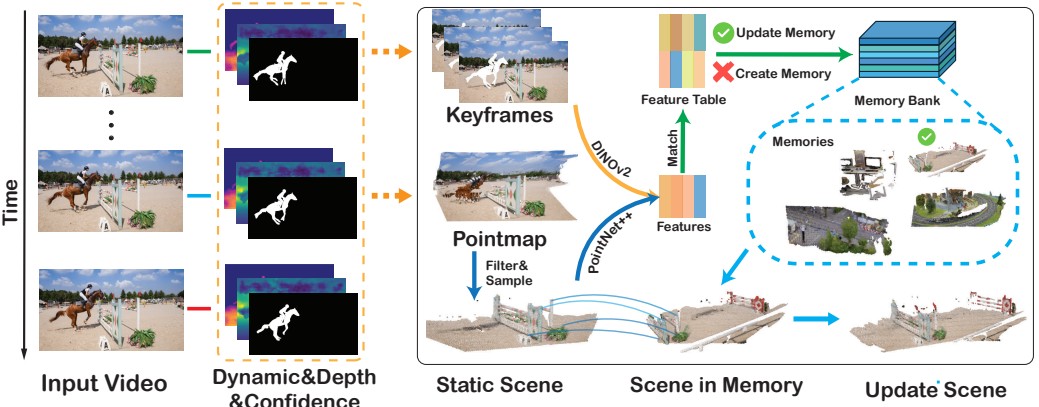

Figure 4: **Cognitive Mapping System.** Given the input video, we estimate per-frame dynamic mask with prior of the depth, confidence, camera pose. DINOv2 and Pointnet++ encode selected static images and static scene into latent features respectively. We then match features with a global feature table, if failed, a new memory slot is created; otherwise the corresponding memory is updated, enabling fast recall, relocalization, and refinement of the current scene.

**Memory Bank Creation.** We construct scalable memory banks using a dual representation strategy that integrates 3D geometric and 2D visual features from static scenes. For 3D features, we filter point clouds to retain only static regions with high confidence, then structure them using an octree hierarchy with adaptive voxel downsampling. These point clouds are encoded with PointNet++ (Qi et al., 2017a), balancing memory efficiency with geometric fidelity. For 2D features, we extract global visual embeddings from static image regions using DINOv2 (Oquab et al., 2023), creating compact representations of each viewpoint. We select representative keyframes by maintaining consistent feature distances between consecutive frames, ensuring balanced visual coverage.

The resulting memory bank is a hierarchical structure, with each scene assigned a unique identifier (map_id). Each map stores a static point cloud $\mathcal{P}_{\text{static}} \in \mathbb{R}^{N_{\text{pts}} \times 3}$ and its associated 2D and 3D features. For efficient retrieval, we implement a global feature table using hash-based approximate nearest neighbor principles and a companion mapping file. This two-tier design separates fast visual search from subsequent loading of geometric data, enabling rapid scene matching and recall.

**Memory Recall and Relocalization.** For new observations, we employ a two-stage approach to determine if the location has been previously mapped. First, we match each query frame's static features against our memory bank by computing L2 distances in feature space. Each successful match casts a vote for its corresponding map, allowing us to identify candidate environments through sequence-level consensus rather than relying on single-frame comparisons.

For the highest-voted candidate map, we conduct geometric verification by aligning its stored point cloud ($\mathcal{P}_{\text{map}}$) with the query sequence's static point cloud ($\mathcal{P}_{\text{query}}$) using ICP. We focus on the absolute count of inlier correspondences and their RMSE, enabling robust matching even with partial scene overlap. This approach effectively distinguishes true relocalization opportunities from visually similar but geometrically distinct environments. Upon successful verification, we obtain the precise 6-DoF camera pose within the map coordinate system, enabling immediate reuse of previously optimized scene representations for subsequent mapping and tracking operations.

**Memory Update.** Following successful relocalization, we update two core components of our memory system: First, we enhance visual recognition by extracting features from new keyframes and adding them to both the global feature table and the matched map's feature set. This enriches visual references for future recognition from multiple viewpoints. Second, we refine geometric representation by transforming the current static point cloud into the matched map's coordinate system using the obtained camera pose. We then merge the aligned data with the existing map and apply consistent voxel downsampling to eliminate redundancy while maintaining resolution quality.

This dual update strategy extends coverage to previously unobserved regions while improving accuracy in overlapping areas. The updated static scene in the memory bank serves three critical functions: enriching the persistent environmental model, refining the current point cloud through

integration of prior knowledge, and providing stronger constraints for subsequent camera trajectory optimization. Each revisit creates a progressive cycle where better recognition enables more precise updates, completing the cognitive loop of storage, recall, and refinement.

### 3.3 CAMERA TRAJECTORY OPTIMIZATION

We propose a factor graph optimization approach to refine camera trajectories, enhancing global geometric consistency through static scene constraints from both current observations and memory.

**Initial Landmark Selection.** Reliable landmark selection is crucial for effective optimization. To ensure high-quality geometric constraints, we extract candidate landmarks exclusively from static regions $(1 - \mathcal{M}_{\text{dyn}})$ with high confidence values. For scenes recognized from memory, we transform stored static points into the current coordinate frame using the alignment transformation computed during memory recall, serving as additional landmarks with established 3D positions, providing stronger geometric constraints. Landmark association employs a two-step verification process with an adaptive threshold $\tau_{dist} = \max(\tau_{min}, d_{scene} \cdot \alpha)$ that scales with scene dimensions.

**Factor Graph Optimization.** Our approach jointly optimizes camera poses $T_i \in SE(3)$ and 3D landmarks $L_j \in \mathbb{R}^3$ by minimizing:

$$X^* = \text{argmin}_X \sum_{f \in F} \|C(X_f)\|^2_{\Sigma_f^{-1}} \tag{3}$$

where $X^* = \{\{T_i\}_{i=0}^{N-1}, \{L_j\}_{j=0}^{M-1}\}$ represents the optimal state. We incorporate three complementary constraints: To establish accurate geometric correspondence, we define projection factors that ensure 3D landmarks align with their 2D observations, as well as a prior factor which anchors the coordinate system:

$$f_{proj}(T_i, L_j) = \pi(T_i, L_j) - \mathbf{z}_{ij}; \quad f_{prior}(T_0) = T_0 \ominus T_0^0 \tag{4}$$

where $\pi$ is the projection function, $\mathbf{z}_{ij}$ is the observed 2D point, $T_0^0$ is the initial camera pose estimate, and $\ominus$ represents the difference in the $SE(3)$ manifold. To encourage physically plausible motion, we enhance trajectory smoothness with inter-frame motion constraints:

$$f_{motion}(T_{i-1}, T_i) = (T_{i-1}^{-1} T_i) \ominus (T_{i-1}^0)^{-1} T_i^0 \tag{5}$$

which naturally penalizes deviations from initial relative transformations between consecutive frames. The complete cost function uses the Huber loss function $\rho$ for robustness:

$$C(X) = |f_{prior}|^2_{\Sigma_{prior}^{-1}} + \sum_{(i,j) \in \mathcal{O}} \rho(|f_{proj}|^2_{\Sigma_{proj}^{-1}}) + \sum_{i=1}^{N-1} |f_{motion}|^2_{\Sigma_{motion}^{-1}} \tag{6}$$

When revisiting familiar scenes, landmarks retrieved from memory are assigned higher confidence by downscaling their projection covariance ($\Sigma_{\text{proj}} \leftarrow \alpha \Sigma_{\text{proj}}, 0 < \alpha < 1$). We minimize $C(X)$ using the Levenberg–Marquardt algorithm with a Huber loss, and apply standard stability enhancements including rotation re-orthogonalization via SVD and adaptive thresholding for outlier rejection.

## 4 EXPERIMENTS

CogniMap3D processes monocular videos of dynamic scenes, providing accurate depth estimates, camera poses, and persistent scene memory. We evaluate against specialized methods for depth estimation, camera tracking and 3D reconstruction, while also demonstrating our system's unique capabilities for scene recognition and memory updates across multiple visits.

**Baselines.** We compare CogniMap3D against state-of-the-art methods that approach different aspects of dynamic scene understanding. Our primary set of baselines includes Spann3R (Wang & Agapito, 2024), MonST3R (Zhang et al., 2024), CUT3R (Wang et al., 2025b), and VGGT (Wang et al., 2025a). MonST3R extends DUSt3R (Wang et al., 2024) to handle dynamic scenes by integrating optical flow analysis for motion segmentation, while Spann3R employs spatial memory mechanisms to process variable-length sequences. CUT3R implements a stateful recurrent model for continuous scene refinement with each new observation. VGGT serves as our foundation model baseline for direct regression of geometric information. While these methods provide strong baselines for depth and pose estimation, our approach uniquely integrates long-term scene memory capabilities for efficient recognition and update of previously visited environments.

Table 1: **Video Depth Evaluation.** We report scale&shift-invariant depth accuracy and FPS. Methods requiring global alignment are marked "GA", while "Optim." and "FF" indicate optimization and online methods.

| Category | Method | Optim. | FF | Sintel Butler et al. (2012) | | BONN Palazzolo et al. (2019) | | KITTI Geiger et al. (2013) | | FPS |
|---|---|---|---|---|---|---|---|---|---|---|
| | | | | Abs Rel ↓ | $\delta$<1.25 ↑ | Abs Rel ↓ | $\delta$<1.25 ↑ | Abs Rel ↓ | $\delta$<1.25 ↑ | |
| Depth Estimation model | Depth-Anything-V2 Yang et al. (2024) | | ✓ | 0.367 | 55.4 | 0.106 | 92.1 | 0.140 | 80.4 | 3.13 |
| | ChronoDepth Shao et al. (2024) | | ✓ | 0.687 | 48.6 | 0.100 | 91.1 | 0.167 | 75.9 | 1.89 |
| | DepthCrafter Hu et al. (2024) | | ✓ | **0.292** | **69.7** | **0.075** | **97.1** | 0.110 | 88.1 | 0.97 |
| Vision Foundation Model | DUSt3R-GA Wang et al. (2024) | ✓ | | 0.531 | 51.2 | 0.156 | 83.1 | 0.135 | 81.8 | 0.76 |
| | MASt3R-GA Leroy et al. (2024) | ✓ | | 0.327 | 59.4 | 0.167 | 78.5 | 0.137 | 83.6 | 0.31 |
| | MonST3R-GA Zhang et al. (2024) | ✓ | | 0.333 | 59.0 | 0.066 | 96.4 | 0.157 | 73.8 | 0.35 |
| | Spann3R Wang & Agapito (2024) | | ✓ | 0.508 | 50.8 | 0.157 | 82.1 | 0.207 | 73.0 | 13.55 |
| | CUT3R Wang et al. (2025b) | | ✓ | 0.454 | 55.7 | 0.074 | 94.5 | 0.106 | 88.7 | 16.58 |
| | VGGT Wang et al. (2025a) | | ✓ | 0.299 | 62.4 | 0.054 | 97.1 | 0.072 | **96.4** | 21.5 |
| | **Ours** | | ✓ | **0.295** | **68.6** | 0.058 | **97.9** | **0.069** | 96.2 | 14.32 |

Table 2: **Quantitative Results of 3D reconstruction.** Evaluation on 7-Scenes dataset shows our approach achieves comparable results to SOTA methods.

| Method | Optim. | FF | 7-Scenes Shotton et al. (2013) | | | | | | FPS |
|---|---|---|---|---|---|---|---|---|---|
| | | | Acc↓ Mean | Acc↓ Med. | Comp↓ Mean | Comp↓ Med. | NC↑ Mean | NC↑ Med. | |
| DUSt3R-GA Wang et al. (2024) | ✓ | | 0.146 | 0.077 | 0.181 | 0.067 | 0.736 | 0.839 | 0.68 |
| MonST3R-GA Zhang et al. (2024) | ✓ | | 0.248 | 0.185 | 0.266 | 0.167 | 0.672 | 0.759 | 0.39 |
| CUT3R Wang et al. (2025b) | | ✓ | 0.126 | 0.047 | 0.154 | **0.031** | 0.727 | 0.834 | 17.0 |
| VGGT Wang et al. (2025a) | | ✓ | 0.088 | **0.040** | 0.092 | 0.040 | **0.784** | **0.888** | 21.5 |
| **Ours** | | ✓ | **0.086** | 0.041 | **0.089** | 0.039 | 0.751 | 0.863 | 14.3 |

## 4.1 VIDEO DEPTH ESTIMATION

**Datasets and Metrics.** Following the benchmarks of previous works (Hu et al., 2024; Zhang et al., 2024; Wang et al., 2025b), our evaluation uses Sintel (Butler et al., 2012), KITTI (Geiger et al., 2013), and Bonn (Palazzolo et al., 2019) datasets, covering synthetic and real-world environments across indoor and outdoor settings. We report absolute relative error (Abs Rel) and percentage of inlier points with $\delta < 1.25$, applying per-sequence scale and shift alignment. We denote "FF" as methods that obtain predictions from a single forward pass of a vision foundation model without pairwise multi-view reconstruction or test-time gradient-based optimization, and "Optim." as methods that perform explicit pairwise reconstructions over image pairs.

**Results.** Table 1 shows CogniMap3D achieves competitive performance across most datasets. Our method outperforms models designed for static scenes like DUSt3R and Spann3R, while matching VGGT and surpassing specialized depth estimation networks like Depth-Anything-V2 (Yang et al., 2024). Though our memory system introduces slight computational overhead, CogniMap3D remains faster than optimization-based approaches, effectively balancing accuracy and efficiency.

## 4.2 3D RECONSTRUCTION AND ANALYSIS

**Qualitative Analysis on Dynamic 3D Reconstruction.** We compare the reconstruction quality of CogniMap3D with MonST3R (Zhang et al., 2024) and CUT3R (Wang et al., 2025b) on the DAVIS (Perazzi et al., 2016) and KITTI (Geiger et al., 2013) datasets, as shown in Fig. 5. MonST3R processes image pairs for dynamic scene reconstruction but lacks global consistency, resulting in visible noise. CUT3R improves upon this with a state-based approach to integrate observations, yet still suffers from error accumulation that causes progressive misalignments across sequences. In contrast, our method builds upon and enhances VGGT's foundation through our multi-stage motion cue analysis, accurately separating dynamic elements from static backgrounds and producing cleaner, more consistent reconstructions across all test scenarios.

**Reconstruction with Memory.** A key advantage of CogniMap3D is its cognitive mapping system, demonstrated in the bottom rows of Fig. 5. When revisiting environments, our system recognizes familiar scenes by matching current observations against stored visual and geometric features, recalls the corresponding memory, and integrates new static information. We visualize this capability by rendering stored memory scenes with higher brightness to distinguish them from newly observed elements. The visualization reveals how subsequent visits integrate additional static elements while maintaining overall scene structure, enabling long-term environmental understanding that more closely mimics human cognitive spatial memory.

**Quantitative Analysis on 3D Reconstruction.** We evaluate our method on the 7-Scenes dataset using accuracy (Acc), completion (Comp), and normal consistency (NC) metrics. Following prior

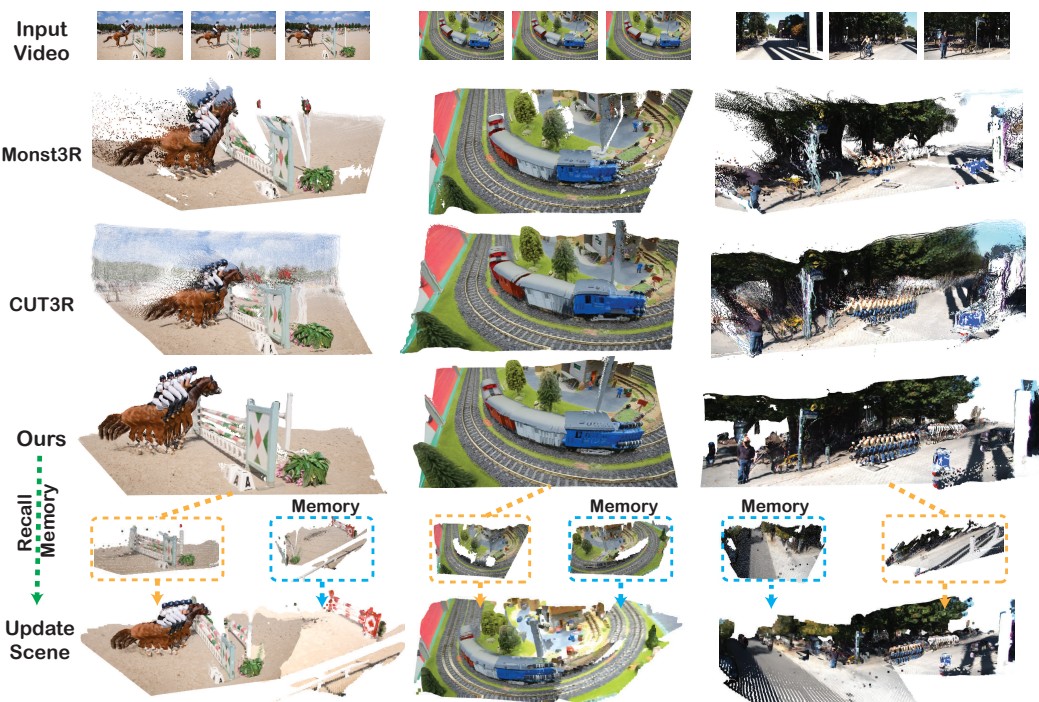

Figure 5: **Qualitative Results of Dynamic 3D Reconstruction.** We compare our method with concurrent works MonST3R (Zhang et al., 2024) and CUT3R (Wang et al., 2025b). Our method achieves cleaner reconstructions with better preservation of both static and dynamic elements. The bottom rows demonstrate CogniMap3D's unique capability to store previous scenes in memory and recall them upon revisitation. We render stored memory scenes with higher brightness to distinguish them from newly observed scene.

works (Wang et al., 2025b; Zhu et al., 2022; Wang et al., 2024; Wang & Agapito, 2024), we evaluate using 3-5 frames per scene. As shown in Table 2, our method achieves comparable results, especially on accuracy and completion metrics.

## 4.3 CAMERA POSE ESTIMATION

**Datasets and Metrics.** To rigorously evaluate CogniMap3D's camera pose estimation capabilities, we employ three complementary datasets that present distinct challenges: Sintel (Butler et al., 2012) with its elaborate dynamic content, TUM-dynamics (Sturm et al., 2012) featuring real-world dynamic scenes with ground truth trajectories, and ScanNet (Dai et al., 2017) to assess generalization to static environments with diverse architectural layouts. Following previous works (Wang et al., 2025b; Chen et al., 2024; Zhang et al., 2024), our quantitative assessment utilizes three key metrics: Absolute Translation Error (ATE) for global trajectory consistency, and Relative Pose Error (RPE) in both translation (RPE trans) and rotation (RPE rot) to measure incremental positional and rotational accuracy over standardized distances.

**Results.** Table 3 presents a comprehensive comparison of camera pose estimation methods across three distinct categories. The first category comprises SLAM-based approaches designed for camera pose estimation, which demonstrate high accuracy but require ground truth camera intrinsics as input. The second category includes optimization-based vision foundation models, which achieve impressive results through scene reconstruction via feature matching but at the cost of computational efficiency. Most notably, CogniMap3D excels in the third category of Feed-Forward methods (FF), achieving superior performance across all datasets with an ATE of 0.176 on Sintel and 0.012 on TUM-dynamics, outperforming competing approaches such as DUSt3R, Spann3R, and CUT3R.

Table 3: **Camera Pose Estimation Evaluation** on Sintel, TUM-dynamic, and ScanNet datasets. We group methods into (I) SLAM-based methods requiring intrinsics, (II) optimization-based VFM methods, and (III) feed-forward VFM methods.

| Category | Method | Sintel Butler et al. (2012) | | | TUM-dynamic Sturm et al. (2012) | | | ScanNet Dai et al. (2017) | | |
|---|---|---|---|---|---|---|---|---|---|---|
| | | ATE ↓ | RPE trans ↓ | RPE rot ↓ | ATE ↓ | RPE trans ↓ | RPE rot ↓ | ATE ↓ | RPE trans ↓ | RPE rot ↓ |
| I | DROID-SLAM Teed & Deng (2021) | 0.175 | 0.084 | _1.912_ | - | - | - | - | - | - |
| | DPVO Teed et al. (2023) | _0.115_ | _0.072_ | 1.975 | - | - | - | - | - | - |
| | LEAP-VO Chen et al. (2024) | **0.089** | **0.066** | **1.250** | 0.068 | 0.008 | 1.686 | 0.070 | 0.018 | 0.535 |
| II | Robust-CVD Kopf et al. (2021) | 0.360 | 0.154 | 3.443 | 0.153 | 0.026 | 3.528 | 0.227 | 0.064 | 7.374 |
| | CasualSAM Zhang et al. (2022) | _0.141_ | **0.035** | **0.615** | _0.071_ | **0.010** | 1.712 | 0.158 | 0.034 | 1.618 |
| | DUSt3R-GA Wang et al. (2024) | 0.417 | 0.250 | 5.796 | 0.083 | 0.017 | 3.567 | 0.081 | 0.028 | 0.784 |
| | MASt3R-GA Duisterhof et al. (2024) | 0.185 | 0.060 | 1.496 | **0.038** | _0.012_ | **0.448** | _0.078_ | _0.020_ | **0.475** |
| | MonST3R-GA Zhang et al. (2024) | **0.111** | _0.044_ | _0.869_ | 0.098 | 0.019 | _0.935_ | **0.077** | **0.018** | _0.529_ |
| III | DUSt3R Wang et al. (2024) | 0.290 | 0.132 | 7.869 | 0.140 | 0.106 | 3.286 | 0.246 | 0.108 | 8.210 |
| | Spann3R Wang & Agapito (2024) | 0.329 | 0.110 | 4.471 | 0.056 | 0.021 | 0.591 | 0.096 | 0.023 | 0.661 |
| | CUT3R Wang et al. (2025b) | 0.213 | **0.066** | 0.621 | 0.046 | _0.015_ | 0.473 | 0.099 | 0.022 | 0.600 |
| | VGGT Wang et al. (2025a) | _0.189_ | 0.069 | **0.529** | _0.028_ | 0.020 | _0.350_ | _0.023_ | _0.015_ | **0.326** |
| | **Ours** | **0.176** | _0.068_ | _0.600_ | **0.012** | **0.010** | **0.311** | **0.019** | **0.011** | _0.331_ |

Table 4: **Memory Recall Analysis.**

| Method | Acc↓ | Comp↓ | NC↑ |
|---|---|---|---|
| MonST3R-GA | 0.248 | 0.266 | 0.672 |
| **Ours** | _0.086_ | _0.089_ | _0.751_ |
| **Ours Update** | **0.082** | **0.085** | **0.789** |

Table 5: **Camera Pose Analysis.**

| Method | ATE↓ | RPE↓ |
|---|---|---|
| Baseline | 0.024 | _0.334_ |
| PnP+RANSAC | 0.025 | 0.510 |
| DPVO | _0.019_ | 0.510 |
| **Ours** | **0.012** | **0.311** |

Table 6: **Memory Size.**

| Number | Accuracy (%) |
|---|---|
| 1 | 100 |
| 50 | 96 |
| 100 | 97 |
| 200 | 97.5 |

## 4.4 ABLATION STUDY

**Memory Recall Analysis.** Our model continuously recalls, updates, and stores 3D scenes in memory. When processing image streams from previously visited environments, it retrieves stored representations to assist current scene understanding. We demonstrate this capability on the 7-Scenes dataset as shown in Table 4. For evaluation, we first initialize the memory bank with a single randomly selected frame from each scene. Leveraging memory recall, our method with updated memory outperforms both baseline methods and our model without memory.

**Camera Pose Methods.** We evaluate multiple methods for stabilizing initial camera poses from VFM. Our baseline is established without camera refinement, memory recall, or any pose adjustments beyond static scene estimation. Tab. 5 indicate that existing methods struggle: PnP+RANSAC (Gao et al., 2003) suffers from poor temporal consistency, while learning-based methods like DPVO (Teed et al., 2023) maintain internal states incompatible with VFM's prior. Our factor graph optimization jointly refines camera extrinsics and landmark positions, reducing trajectory error and maintaining rotation precision.

**Memory Matching.** We evaluate CogniMap3D's memory matching on DAVIS (Perazzi et al., 2016) by dividing 50 scenes into thirds and performing 200 pairwise matches between segments. As Table 6 shows, our system maintains high accuracy in increasing memory sizes, demonstrating robust feature-based matching for effective scene recognition and camera pose refinement.

## 5 CONCLUSION

We presented CogniMap3D, a bioinspired framework for dynamic scene understanding that emulates key aspects of human cognitive processing through three complementary capabilities: a multi-stage motion cue framework that progressively distinguishes dynamic objects from static backgrounds, a cognitive mapping system that creates and maintains persistent environmental memory, and a camera pose refinement strategy that establishes reliable coordinate frames through factor graph optimization. Our comprehensive evaluation demonstrates superior performance in depth estimation, camera pose estimation, and 3D reconstruction across diverse datasets.

## ETHICS STATEMENT

Our research targets scientific exploration of dynamic scene understanding and 3D scene memory systems, with potential benefits for autonomous navigation, augmented reality, and assistive technologies. We also recognize risks such as inadvertent privacy breaches and misuse for surveillance or tracking. Any deployment should follow transparent usage protocols and comply with applicable privacy regulations and ethical guidelines. This work is developed for academic research purposes, and we discourage applications that could infringe individual privacy.

## REPRODUCIBILITY STATEMENT

We aim to make COGNIMAP3D fully reproducible. The main paper (Sec. 3) specifies the end-to-end pipeline (motion cues, memory, factor-graph refinement) and evaluation protocols (ATE, RPE trans/rot). The supplementary material (Sec. B) provides implementation details, including model backbones and third-party components, all hyperparameters, random seeds, dataset splits and preprocessing, and configuration specifications. The code repository is linked on the first page.

## ACKNOWLEDGEMENT

This research is supported by NSF IIS-2525840, CNS-2432533, CNS-2432534, ECCS-2514574, NIH 1RF1MH133764-01 and Cisco Research unrestricted gift. This article solely reflects opinions and conclusions of authors and not funding agencies.

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

## Appendix: Cognitive 3D Mapping and Rapid Retrieval

The appendix provides limitations and future directions (Sec. A), detailed implementation specifications including model backbones, and configurations (Sec. B), visualizations of our multi-stage motion cue pipeline (Sec. C), and supplementary results covering robustness analysis, large-scale reconstruction, ablation studies, qualitative comparisons, and memory footprint evaluation (Sec. D).

## A    LIMITATION

Although we have successfully implemented a memory bank system for 3D scene recall with a high matching success rate when revisiting similar environments, mismatching incidents occasionally introduce noise into the system. Further optimization of our matching algorithm is required to ensure more stable and consistent performance across varied environmental conditions. Additionally, our point cloud registration accuracy is contingent upon the quality of static scene representation, which can lead to potential misalignments between point clouds captured from identical scenes under different conditions or perspectives. Factors such as lighting variations, occlusions, and viewpoint changes can impact registration accuracy. We plan to address these limitations through robust feature extraction techniques and adaptive registration algorithms in our future research initiatives.

## B    IMPLEMENTATION DETAILS

We incorporate VGGT (Wang et al., 2025a) to provide initial depth estimation and camera pose priors for our system. Our method is fully implemented in PyTorch and all experiments are conducted on an NVIDIA A6000 GPU with 48GB VRAM. For optical flow computation between consecutive frames, we employ RAFT (Teed & Deng, 2020) with a resolution of $840 \times 480$ pixels, while feature matching across non-consecutive images is performed using LoFTR (Sun et al., 2021) with shared self and cross attention mechanisms. To efficiently encode downsampled point clouds into compact 3D feature representations, we implement a modified version of PointNet++ (Qi et al., 2017b) with three set abstraction layers and feature dimensions of 128, 256, and 512 respectively. For visual feature extraction, we utilize the backbone architecture of VGGT, specifically leveraging DINOv2 with its self-supervised training paradigm to encode keyframes into rich 3D feature representations with a dimensionality of 1024. Our implementation of Perspective-n-Point (PnP) and Random Sample Consensus (RANSAC) algorithms closely follows the methodology outlined in (Gao et al., 2003), with an inlier threshold of 2.0 pixels and a maximum of 1000 iterations. Finally, we implement DPVO (Teed et al., 2023) using the same camera intrinsic parameters as VGGT to maintain consistency in our visual odometry pipeline.

For each incoming frame, we perform a full VGGT forward pass to obtain depth and camera pose, and reuse its intermediate DINOv2 features as 2D descriptors, so no additional DINOv2 model is invoked. RAFT and SAM2 are both applied at a downsampled resolution to estimate optical flow and track masks. The memory-related back-end is invoked every 20 frames rather than at every frame, since neighboring frames observe very similar content. At these keyframes, we first perform 2D memory recall over the feature bank; only when high-confidence candidates are found do we activate 3D geometric validation and a factor-graph update. The 3D validation uses PointNet++ features and ICP on downsampled sparse point clouds of retrieved static landmarks before fusing them into the global map.

## C    VISUALIZATION OF MULTI-STAGE MOTION CUE

Our multi-stage motion cue framework processes video sequences to accurately segment dynamic objects in complex scenes with moving cameras. As illustrated in Fig. 6 and Fig. 7, this progressive refinement approach consists of four key stages:

First, we compute optical flow between consecutive frames using RAFT (Teed & Deng, 2020), capturing motion information throughout the scene. The resulting flow fields (shown in the top-left of each figure) contain motion vectors for both dynamic objects and background affected by camera movement.

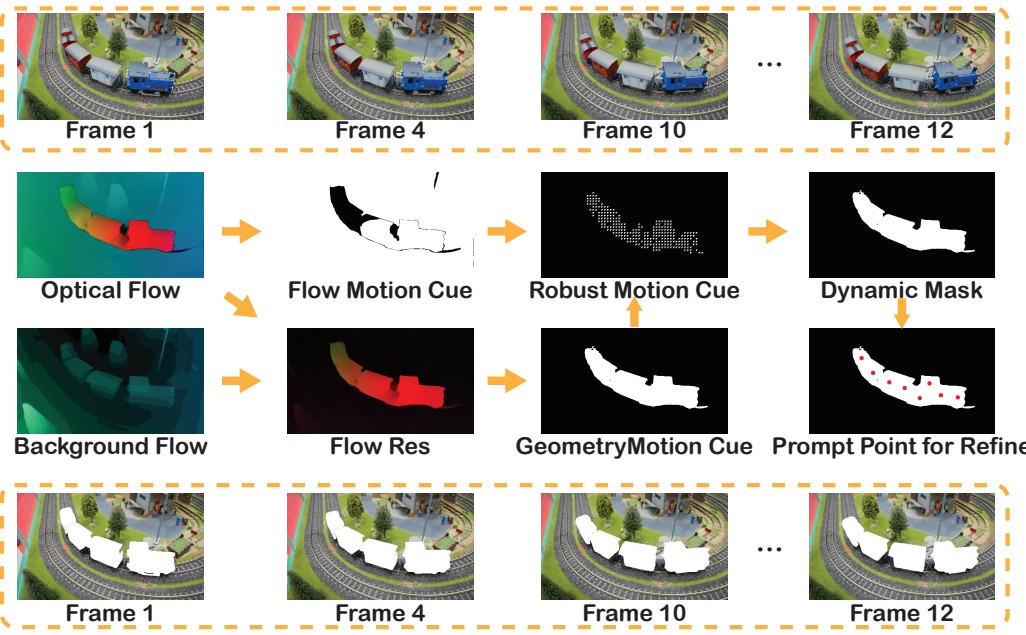

Figure 6: **Motion Cue Process of the Train Scene in DAVIS dataset.**

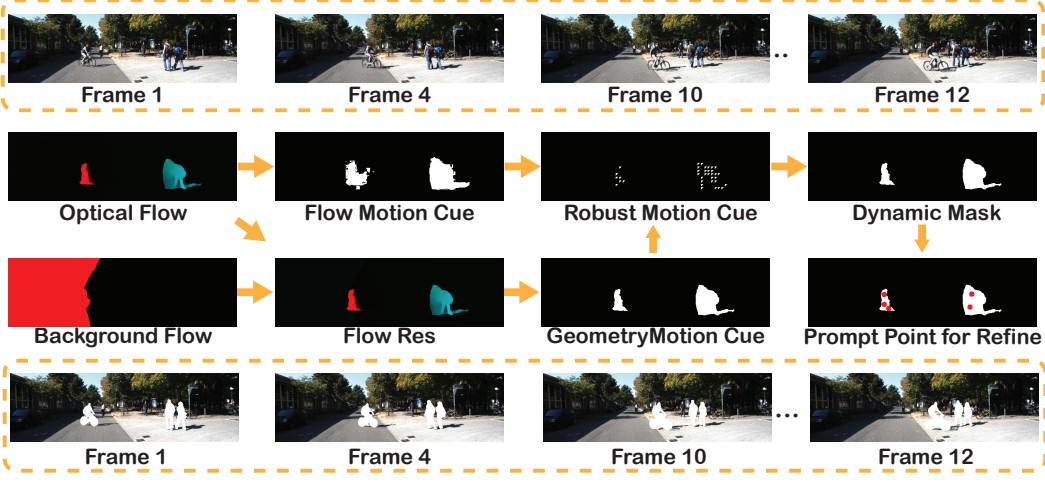

Figure 7: **Motion Cue Process of the Street Scene in KITTI dataset.**

Second, we implement two parallel motion cue extraction processes. The Flow Motion Cue isolates potential dynamic regions by partitioning the optical flow field into distinct components via Gaussian Mixture Model clustering, excluding components with minimal motion magnitude. Simultaneously, we compute the Background Flow by transforming pointmaps through relative camera poses and projecting them onto image planes.

Third, we generate the Flow Residual by subtracting the expected background flow from the observed optical flow, effectively highlighting motion caused exclusively by dynamic objects. This residual is thresholded using Otsu's method to produce the Geometry Motion Cue, which more accurately distinguishes genuine object motion from camera-induced apparent motion.

Finally, we derive the Robust Motion Cue by analyzing keypoint correspondences between frames and identifying outlier displacements. This refined information produces our Dynamic Mask, with

red dots indicating prompt points for further refinement using SAM2. The bottom row of each figure demonstrates how our approach maintains consistent dynamic object segmentation across multiple frames (1, 4, 10, and 12), effectively handling diverse scenarios ranging from model train sets with predictable motion patterns to real-world street scenes with cyclists and pedestrians.

# D  SUPPLEMENTARY RESULTS AND ANALYSES

## D.1  ROBUSTNESS TO INITIALIZATION ERROR

To further assess robustness to initialization errors, we add zero-mean Gaussian noise with rotation standard deviation $\sigma_R$ and translation standard deviation $\sigma_t$ to the initial poses, and compare trajectory errors on the TUM-dynamic dataset. The comparison between the VGGT baseline and our method under different levels of pose perturbation is reported in Table 7.

Table 7: **Robustness to noisy pose initialization on TUM-dynamic.**

| Method | Noise level ($\sigma_R/\sigma_t$) | ATE ↓ | RPE trans ↓ | RPE rot ↓ | Perturbation level |
|---|---|---|---|---|---|
| VGGT | – | 0.028 | 0.020 | 0.350 | – |
| Ours | 0.0°/0.00 m | 0.012 | 0.010 | 0.311 | none |
| Ours | 1.0°/0.01 m | 0.013 | 0.011 | 0.318 | slight |
| Ours | 3.0°/0.03 m | 0.015 | 0.012 | 0.331 | moderate |
| Ours | 5.0°/0.05 m | 0.019 | 0.014 | 0.343 | severe |

Under slight and moderate perturbations, CogniMap3D maintains performance close to the clean initialization. Even with severe perturbations of 5°, it still outperforms the VGGT baseline, indicating robustness to reasonably large pose initialization noise.

## D.2  RECONSTRUCTION IN LARGE-SCALE OUTDOOR SCENES

CogniMap3D is designed to maintain a stable memory of static structure that can be efficiently reused when scenes are revisited, so that repeated long-term visits can enhance geometric consistency instead of accumulating drift. To evaluate this ability in more challenging, open-world settings, we consider KITTI, a standard large-scale outdoor benchmark with street and highway driving sequences, containing repeated viewpoints, moving vehicles, pedestrians, and strong viewpoint and illumination changes.

Table 8: **3D reconstruction on KITTI (outdoor driving sequences).**

| Method | Acc↓ Mean | Acc↓ Med. | Comp↓ Mean | Comp↓ Med. | NC↑ Mean | NC↑ Med. |
|---|---|---|---|---|---|---|
| CUT3R | 0.089 | 0.058 | 0.108 | 0.078 | 0.895 | 0.912 |
| VGGT | 0.071 | 0.045 | 0.089 | 0.061 | 0.913 | 0.935 |
| Ours | **0.052** | **0.036** | **0.073** | **0.049** | **0.942** | **0.951** |

As shown in Table 8, CogniMap3D attains lower accuracy and completeness errors and higher normal consistency than CUT3R and VGGT on KITTI. Within this benchmark, these results indicate that the proposed memory mechanism can be beneficial not only on smaller indoor settings evaluated in the main paper but also in large-scale, long-term outdoor sequences with dynamic objects and revisits.

## D.3  ABLATION OF 2D, 3D FEATURES, AND GEOMETRIC VALIDATION

In CogniMap3D, memory recall and validation are implemented as a three-stage pipeline rather than as independent modules. First, 2D visual features are used for coarse candidate retrieval from the memory bank. Second, the candidates are refined using 3D geometric features. Finally, an ICP-based 3D geometric validation decides whether a candidate is accepted and fused into the map, or rejected so that a new memory entry is created.

To evaluate the contribution of components, we perform an ablation study on DAVIS dataset under the scene-matching setting of Table 6 with a memory bank of size 200. We compare: (i) *2D-only*,

Table 9: **Memory ablation on DAVIS (scene matching).**

| Variant | 3D features | ICP validation | Accuracy ↑ (%) | Throughput ↑ (q/s) |
|---|---|---|---|---|
| 2D-only | ✗ | ✗ | 93.5 | 18.2 |
| 2D+3D, no ICP | ✓ | ✗ | 96.8 | 10.5 |
| Full (2D+3D+ICP) | ✓ | ✓ | 97.5 | 6.8 |

which directly accepts the top-1 match from coarse candidates; (ii) *2D+3D, no ICP*, which augments with 3D features but omits geometric validation; and (iii) *Full*, which uses the complete 2D+3D+ICP pipeline. We report scene recall accuracy and matching throughput (queries per second):

As shown in Table 9, 2D-only matching offers the highest throughput but lower accuracy. Adding 3D features improves recall with a moderate reduction in speed. The full three-stage variant further increases accuracy at the cost of additional computation, illustrating the trade-off between robustness and efficiency in the memory recall module.

### D.4 QUALITATIVE COMPARISON ON DYNAMIC SCENES

To illustrate the performance of CogniMap3D on dynamic scenes, we provide a qualitative comparison with VGGT on a KITTI sequence in Fig. 8. The scene contains a moving cyclist and pedestrians, both of which induce strong motion relative to the static background.

In the VGGT reconstruction, the moving cyclist appears with smeared and partially duplicated geometry, and the wheels and upper body are less clearly delineated. The pedestrians on the right also exhibit blurrier shapes and less coherent structure. In contrast, the CogniMap3D reconstruction shows a more complete and visually consistent shape for the cyclist, including better-defined wheels and torso, and provides sharper, more coherent geometry for the pedestrians. In this example, these visual differences suggest that the proposed dynamic-mask pipeline helps produce cleaner reconstructions around moving objects while preserving the static background.

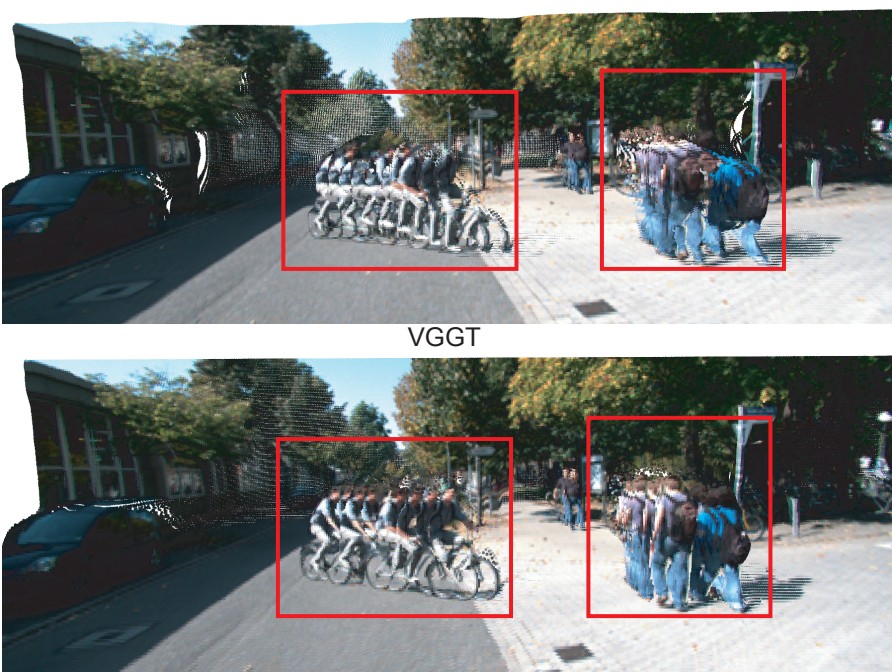

Figure 8: **Qualitative comparison on a KITTI sequence.** We compare reconstructed point clouds rendered over the input image for VGGT (top) and CogniMap3D (bottom). The red boxes highlight regions with moving area (cyclist and pedestrians). In this example, CogniMap3D yields more complete and visually coherent reconstructions of the moving regions.

## D.5 MEMORY FOOTPRINT

We examine the memory footprint of CogniMap3D. In our implementation, the persistent scene memory is stored on the CPU as downsampled static landmarks with compact feature descriptors, indexed by a hash table. This memory is maintained in host RAM and can grow with the number of revisited scenes.

At inference time, only the subset of landmarks that are relevant to the current frame is transferred to the GPU for recall and optimization. As a result, GPU usage is largely dominated by the VGGT forward pass rather than by the size of the stored memory. Measured on our setup, VGGT alone uses about 11.7 GB of GPU memory, while CogniMap3D uses about 11.9 GB due to the additional memory and optimization modules. This small increase indicates that the GPU memory consumption does not grow proportionally with the total size of the scene memory and remains close to that of running VGGT alone.

## D.6 EFFECT OF MEMORY ON LONG-SEQUENCE POSE ESTIMATION

To analyze how the proposed memory mechanism affects camera pose estimation on long trajectories with natural revisits, we evaluate CogniMap3D on several KITTI driving sequences of extended length. We compare three variants: (i) the VGGT backbone, which provides the initial depth and pose estimates; (ii) CogniMap3D without memory, which applies motion cues and a static-only factor graph but does not recall or fuse past scenes; and (iii) the full CogniMap3D with memory-enabled recall and map fusion. We report absolute trajectory error (ATE), translational relative pose error (RPE trans), and rotational relative pose error (RPE rot), averaged over the selected sequences.

Table 10: **Long-sequence pose evaluation on KITTI.**

| Method | Memory | ATE ↓ | RPE trans ↓ | RPE rot ↓ |
|---|---|---|---|---|
| VGGT | ✗ | 0.092 | 0.034 | 0.421 |
| Ours w/o mem | ✗ | 0.068 | 0.026 | 0.367 |
| Ours w/ mem | ✓ | 0.060 | 0.023 | 0.352 |

As shown in Table 10, both CogniMap3D variants yield lower ATE and RPE than VGGT on these long outdoor trajectories, and enabling memory provides further improvements on all three pose metrics. Within this evaluation, the results suggest that, on these sequences, the memory module can provide additional benefits for pose accuracy and consistency beyond the VGGT backbone and the no-memory variant.

# E LLM USAGE

Large language models (LLMs) were used only to aid in wording and polishing the writing. They were not involved in the research design, methodology, experiments, or analysis.

