# OpenReview forum: "CogniMap3D: Cognitive 3D Mapping and Rapid Retrieval"
_ICLR.cc/2026/Conference — ICLR 2026 Poster_

### Official Review · Reviewer_tg31 · 2025-10-26

**Soundness:** 3
**Presentation:** 3
**Contribution:** 3
**Rating:** 6
**Confidence:** 3

**Summary:**

This paper introduces CogniMap3D, a bio-inspired framework for dynamic 3D scene understanding that mimics human cognition by maintaining a persistent memory of static environments. The core innovation lies in integrating three capabilities: a multi-stage motion detection system to identify dynamic objects, a cognitive mapping system for storing and recalling static scenes across multiple visits, and a factor graph optimization for refining camera poses. Evaluations show state-of-the-art performance in depth estimation, pose reconstruction, and 3D mapping, enabling continuous scene understanding over extended periods and repeated visits to the same location.

**Strengths:**

1. The cognitive 3D mapping is intuitive and novel, which effectively decouple static and dynamic parts of the scene.

2. The writing quality, experimental results and visualization are good.

**Weaknesses:**

1. The performance of CogniMap3D is close to VGGT in all experiments, but the inference speed is much lower than VGGT. More discussion on the performance is required.

2. What is the memory footprint of the proposed method. Does the proposed memory mechanism require high GPU memory?

**Questions:**

In Table 1, the proposed method belongs to 'FF' category. However, different from CUT3r and VGGT which are eng-to-end, the proposed method requires an optimization process to refine camera trajectory. Is the optimization feed-forward?

---

> ### Author Response · Authors · 2025-11-22
>
> #  Weakness 1 Discussion on Performance and Runtime
> We agree with the reviewer that CogniMap3D introduces additional computation compared to VGGT, and this leads to a lower FPS than a single VGGT forward pass. Our goal, however, is not to replace VGGT with a faster backbone, but to build a full streaming system on top of it that explicitly handles dynamics, long-term revisits, and global consistency.
>
> Concretely, CogniMap3D adds: (i) a multi-stage motion-cue module that explicitly segments dynamic and unreliable regions instead of implicitly relying on VFM predictions; (ii) a memory mechanism that supports fast content-based recall, retrieval, and update of static geometry to assist revisits and long-term reconstruction; and (iii) a static-only factor-graph optimization that jointly refines poses and landmarks in vetted static regions.
>
> As shown in Tables 1–3 in the main paper, although these strategies reduce the FPS compared to VGGT, they consistently improve overall camera pose, depth estimation, and 3D reconstruction quality. Our method runs at around 14 FPS, which remains competitive compared to pairwise reconstruction methods such as MonST3R, typically running below 1 FPS.
>
> # Weakness 2 Memory footprint and GPU requirements
>
> Thank you for raising this important point. The proposed memory mechanism has a moderate CPU memory footprint and only a small overhead on GPU memory. In our implementation, VGGT alone uses about 11.7 GB of GPU memory, while CogniMap3D uses about 11.9 GB due to the additional memory and optimization modules.
>
> The persistent scene memory is stored in host RAM as downsampled static landmarks with compact feature descriptors, indexed by a CPU-side hash table. At inference time, only the subset of landmarks relevant to the current frame is transferred to the GPU, so GPU usage does not grow linearly with the size of the memory and remains dominated by the VGGT forward pass. We have provided additional details on the memory footprint and GPU usage in Appendix E.5.
>
> # Question 1 Clarification of the “FF” Category
>
> We thank the reviewer for pointing out this potential confusion. In Table 1, the “FF” category refers to methods that obtain their predictions with a single forward pass of a vision foundation model, without explicit pairwise reconstruction between many image pairs and without any test-time gradient-based optimization. All methods in this category, such as CUT3R, VGGT, and CogniMap3D, achieve real-time inference above 10 FPS.
>
> By contrast, the methods listed in the “Optim.” columns, such as DUSt3R-GA, MASt3R-GA, and MonST3R-GA, first perform explicit pairwise reconstructions over many image pairs, leading to sub-FPS runtimes.
>
> CogniMap3D does include an additional factor-graph optimization, but this step is deliberately lightweight and low-frequency. In practice it introduces only a small overhead compared to the VFM forward pass, and our reported speed of 14.32 FPS is computed as the total number of frames divided by the total wall-clock time of the complete pipeline, including this refinement. We have further clarified the definitions of the “FF” and “Optim.” categories in Section 4.1.

---

> > ### Comment · Reviewer_tg31 · 2025-11-25
> >
> > Thanks the authors for their reply. It solves most of my concerns. But I still think the performance improvement over VGGT is marginal, according to Table 1,2,3. On most of the metrics, the improvement is within 0.5%, sometimes even lower than VGGT. Any explanation on the results?

---

> > > ### Author Response · Authors · 2025-11-25
> > >
> > > We thank the reviewer for the insightful observation regarding the results in Tables 1–3. We would like to provide the context of the benchmarks used in these experiments, and conduct additional evaluation on large-scale outdoor scenes to further demonstrate the superiority of our CogniMap3D.
> > >
> > >
> > > **1. Context of the Benchmarks in Tables 1-3.**
> > >
> > > The performance comparison observed in Tables 1-3 may reflect the evaluation settings: ScanNet and 7-Scenes are near-static indoor scenes, while BONN and Sintel are dynamic but involve relatively small-scale scenes. In these scenarios, VGGT already performs strongly, and CogniMap3D is designed to improve the reconstruction by explicitly handling dynamic objects with a memory mechanism, which is most beneficial in larger-scale, dynamic environments. Among the original benchmarks, TUM-dynamic already shows a clear and consistent improvement in pose accuracy over VGGT in Table 3.
> > >
> > > | Method | ATE ↓ | RPE trans ↓ | RPE rot ↓ |
> > > |--------------------------------------|-----------|------------------|----------------|
> > > | CUT3R     | 0.046     | 0.015            | 0.473          |
> > > | VGGT      | 0.028     | 0.020            | 0.350          |
> > > | Ours                              | **0.012** | **0.010**        | **0.311**      |
> > >
> > >
> > >
> > > **2. Large-scale outdoor evaluation.**
> > >
> > > To better assess performance in the large-scale scenarios highlighted by the reviewers, we conducted additional experiments on KITTI, a standard large-scale outdoor benchmark with street and highway driving sequences and moving vehicles and pedestrians.
> > >
> > > **2.1 Experiment on 3D reconstruction.** We conducted a supplementary experiment for 3D reconstruction on the KITTI dataset. As shown in the table below, CogniMap3D achieves better 3D reconstruction results than the baselines across most metrics. See Appendix E.2 for details.
> > >
> > > | Method | Acc↓ Mean | Acc↓ Med. | Comp↓ Mean | Comp↓ Med. | NC↑ Mean | NC↑ Med. |
> > > |--------|-----------|-----------|------------|------------|----------|----------|
> > > | CUT3R  | 0.089     | 0.058     | 0.108      | 0.078      | 0.895     | 0.912     |
> > > | VGGT   | 0.071     | 0.045     | 0.089      | 0.061      | 0.913     | 0.935     |
> > > | Ours   | **0.052**     | **0.036**     | **0.073**      | **0.049**      | **0.942**     | **0.951**     |
> > >
> > > **2.2 Evaluation on pose estimation.**  We additionally evaluated camera pose estimation on the KITTI dataset. We compared (i) the VGGT backbone, (ii) CogniMap3D without memory, which uses motion cues and a static-only factor graph but does not recall or fuse past scenes, and (iii) the full CogniMap3D with memory enabled.  Both CogniMap3D variants reduce ATE and RPE compared to VGGT on long outdoor scenes, and enabling memory further improves all pose metrics. See Appendix E.6 for details.
> > >
> > >
> > > | Method        | Memory | ATE ↓ | RPE trans ↓ | RPE rot ↓ |
> > > |--------------|--------|-------|-------------|-----------|
> > > | VGGT         | ✗      | 0.092 | 0.034       | 0.421     |
> > > | Ours w/o mem | ✗      | 0.068 | 0.026       | 0.367     |
> > > | Ours w/ mem  | ✓      | 0.060 | 0.023       | 0.352 |
> > >
> > > **2.3 Visualization Comparison.** We further provided a qualitative comparison with VGGT in Figure 8 (Appendix E.4), which shows reduced ghosting artifacts and more complete reconstructions in dynamic regions.
> > >
> > > Taken together, these results indicate that while CogniMap3D is broadly comparable to VGGT on standard short indoor benchmarks, its advantages become clearer in dynamic, large-scale, and long-horizon scenarios where the memory and motion-cue mechanisms provide the most benefit.

---

> > > > ### Comment · Reviewer_tg31 · 2025-11-27
> > > >
> > > > That's a clear explanation. I think my concerns are well solved. It is recommended to include these experiments in the revised version. Existing experiments may be too limited to show the advantages of proposed method.

---

> > > > > ### Author Response · Authors · 2025-11-27
> > > > >
> > > > > We appreciate the reviewer's constructive feedback and are glad that our clarifications addressed the concerns. We will make these experiments more prominent in the revised version to better demonstrate the advantages of the proposed method.

---

### Official Review · Reviewer_RgAW · 2025-10-31

**Soundness:** 3
**Presentation:** 3
**Contribution:** 2
**Rating:** 4
**Confidence:** 4

**Summary:**

This paper proposes a feed-forward 3D network for dynamic scenes. It separates static and dynamic regions using dynamic mask and introduces a memory system. The results showed more stable camera pose and depth estimation under motion compared with baselines.

Contributions:

1. A multi-stage motion-cue module that progressively refines the dynamic mask using geometry and optical flow, which is more stable than the MonST3R masks.
2. A memory system for mapping, recall, and update, enabling robust relocalization.

**Strengths:**

1. This paper provides clear writing and figures. The dynamic mask pipeline is easy to follow.
2. Geometry and flow cues plus global mean move refine masks and stabilize tracking. The results showed that 3D reconstruction is more stable than baselines in metrics and visualization.
3. Using 2D/3D features supports fast indexing, and voting + ICP verifies matches. The memory further supplies constraints that help the global optimization of camera trajectories.

**Weaknesses:**

1. VGGT comparison gap. The model initializes pose and depth with VGGT, yet lacks direct, controlled comparisons against VGGT in the visualizations; reconstruction metrics are also close. Given VGGT is not specifically optimized for dynamic scenes, should include more dynamic motion benchmarks and report VGGT vs. your method to substantiate the value of dynamic-mask extraction.
2. Lack evidence for the memory module. While the paper states that memory provides stronger constraints for trajectory optimization, the ablation focuses on 3D reconstruction only. The specific contribution of memory to pose accuracy/consistency remains under-substantiated. Consider testing memory on longer sequences and evaluating improvements on tasks/settings where VGGT might struggle (e.g., long-horizon drift, relocalization).

**Questions:**

1. Can you provide **head-to-head comparisons with raw VGGT outputs** on scenes with **more dynamic objects**, and analyze the module’s concrete gains on moving regions?
2. Can you report **memory-induced improvements** across **additional benchmarks** (eg. pose esitimation)?
3. Can you **evaluate memory on longer sequences** to demonstrate practicality and its effect on drift/relocalization beyond VGGT?

---

> ### Author Response · Authors · 2025-11-22
>
> # Weakness 1 & Question 1 VGGT comparison on dynamic scenes
>
> We thank the reviewer for suggesting head-to-head visual comparisons with VGGT on dynamic scenes. In addition to the quantitative results, we have added side-by-side visualization on KITTI dataset in the appendix E.5.
>
> As shown in Fig. 8, VGGT can yield smeared or partially duplicated traces of the moving actors and less consistent geometry in these regions. In contrast, our dynamic-mask pipeline better preserves the static background and produces more complete, visually coherent reconstructions of the moving objects, suggesting clearer geometry in these dynamic areas.
>
> # Weakness 2 & Question 2 Ablation of Memory Mechanism for Camera Pose
>
> We thank the reviewer for highlighting the memory module’s contribution to camera pose estimation. We add an evaluation for camera pose estimation on several long KITTI driving sequences with natural revisits. We compare (i) the VGGT backbone, (ii) CogniMap3D without memory, which uses motion cues and a static-only factor graph but does not recall or fuse past scenes, and (iii) the full CogniMap3D with memory enabled. We report ATE, translational RPE, and rotational RPE averaged over these sequences.
>
> **Table: Long-sequence pose evaluation on KITTI.**
>
> | Method        | Memory | ATE ↓ | RPE trans ↓ | RPE rot ↓ |
> |--------------|--------|-------|-------------|-----------|
> | VGGT         | ✗      | 0.092 | 0.034       | 0.421     |
> | Ours w/o mem | ✗      | 0.068 | 0.026       | 0.367     |
> | Ours w/ mem  | ✓      | 0.060 | 0.023       | 0.352     |
>
> Both CogniMap3D variants reduce ATE and RPE compared to VGGT on these long outdoor trajectories, and enabling memory further improves all pose metrics. In Table 4, we also evaluate the effect of the memory module on 3D reconstruction. Taken together, these results suggest that the memory module contributes positively to pose accuracy, consistency, and reconstruction quality beyond the VGGT backbone and the no-memory variant. We have included this ablation in Appendix E.6.
>
> # Weakness 2 & Question 3 Evaluation on long sequences
>
> We thank the reviewer for suggesting an evaluation on long sequences. To probe this regime, we conduct an additional experiment on KITTI, a standard outdoor benchmark with street and highway driving sequences that include moving vehicles, pedestrians, and illumination changes, providing challenging long trajectories for assessing the effect of our method.
>
> We evaluate CUT3R, VGGT, and CogniMap3D on KITTI using standard 3D reconstruction metrics: accuracy (Acc), completeness (Comp), and normalized completeness (NC):
>
> **Table: 3D reconstruction on KITTI (outdoor driving sequences).**
> | Method | Acc↓ Mean | Acc↓ Med. | Comp↓ Mean | Comp↓ Med. | NC↑ Mean | NC↑ Med. |
> |--------|-----------|-----------|------------|------------|----------|----------|
> | CUT3R  | 0.089     | 0.058     | 0.108      | 0.078      | 0.895     | 0.912     |
> | VGGT   | 0.071     | 0.045     | 0.089      | 0.061      | 0.913     | 0.935     |
> | Ours   | 0.052     | 0.036     | 0.073      | 0.049      | 0.942     | 0.951     |
>
> As shown in the table, CogniMap3D outperforms CUT3R and VGGT on KITTI in most metrics, suggesting that our framework remains effective not only on small-scale indoor scenes but also on large-scale outdoor sequences with dynamic objects and longer time horizons, where it can help mitigate drift and improves reconstruction quality. We have extended our evaluation of reconstruction in large-scale outdoor scenes in the supplementary analysis in Appendix E.2.

---

> ### Author Response · Authors · 2025-11-25
>
> We thank the reviewer for the insightful suggestions. We have provided supplementary experiments and detailed analysis, including VGGT comparisons, memory ablation for pose estimation, and long-sequence evaluation on KITTI. We would be grateful for the reviewer's feedback on whether these have addressed the concerns. We are happy to answer any further questions or clarify any points further.

---

### Official Review · Reviewer_rprw · 2025-11-01

**Soundness:** 2
**Presentation:** 3
**Contribution:** 2
**Rating:** 4
**Confidence:** 3

**Summary:**

This paper presents CogniMap3D, a biologically inspired framework for the problem of understanding and reconstructing dynamic 3D scenes from monocular videos. The core contribution is the design of an integrated pipeline, including a multi-stage motion cue separation module, a cognitive mapping system, and a camera trajectory optimization strategy. The cognitive system is capable of creating, storing, recalling, and updating persistent "memories" of static scenes and is designed to mimic the spatial cognitive ability of humans to revisit familiar environments. Experimental results on several standard datasets such as Sintel, KITTI, TUM-dynamics, and 7-Scenes show that the framework achieves competitive performance on multiple tasks.

**Strengths:**

1. Systematically introducing the concept of "cognitive memory" into dynamic scene reconstruction is a novel and visionary attempt, which directly addresses the key challenge of transitioning from processing isolated video clips to achieving long-term, persistent environmental perception.
2. The paper is clearly articulated, effectively conveying its complex system architecture and core ideas through high-quality illustrations, enabling readers to clearly understand its workflow and contributions.

**Weaknesses:**

1. Although 14.32 FPS is reported in Table 1, this speed is quite amazing considering the complexity of the whole system (integrating VGGT, RAFT, DINOv2, PointNet++, SAM2, etc.). It is recommended that the authors more clearly state which modules are covered by this FPS test.
2. The multi-stage, cascaded architecture of the framework raises a key concern that small biases in upstream modules may be amplified later, suggesting that the authors briefly discuss the robustness of the system to initial estimation errors.
3. The paper claims that its key advantage is to handle long-term sequences and scene revisits, but the relevant quantitative evaluation is only carried out on small-scale relocation datasets such as 7-Scenes. This framework lacks validation in larger scale and more challenging long-term real environments.
4. Ablation of memory systems has not been adequately studied, and it would be more compelling to provide a more nuanced analysis of what contributions 2D visual features and 3D geometric features make in scene recall and validation.

**Questions:**

1. With regard to the reported 14.32 FPS, the authors should clarify exactly what modules are included in the metric and clarify whether it covers the complete, computationally intensive back-end flow, from memory retrieval and geometry validation to factor graph optimization.
2. The authors need to supplement the discussion on the robustness of the cascaded framework under initial estimation errors. If the system design includes a specific mechanism to mitigate this error propagation, the principle needs to be explicitly stated.
3. The authors should conduct experiments to quantify the specific contribution of the 3D geometry validation link. For example, how does the system's performance change after using only 2D visual features for scene recall and relocation (i.e., removing 3D validation)?

---

> ### Author Response · Authors · 2025-11-22
>
> # Weakness 1 & Question 1 Clarification of the FPS measurement
> We thank the reviewer for highlighting the need to clarify the FPS measurement. The reported 14.32 FPS is measured over the full CogniMap3D pipeline. Specifically, for each incoming frame, we run the full VGGT forward pass for depth and pose and reuse its intermediate DINOv2 features as 2D descriptors (i.e., VGGT’s full cost is included and we do not run an additional DINOv2 model). We also run RAFT and SAM2 at downsampled resolution to obtain motion cues and track masks.
>
> The memory-related back-end flow is invoked every 20 frames rather than at every frame, since neighboring frames see very similar content. At these keyframes, we first perform 2D memory recall over the feature bank; only when high-confidence candidates are found do we activate 3D geometry validation and a factor-graph update. The 3D validation uses PointNet++ features and ICP on downsampled sparse point clouds of retrieved static landmarks.
>
> The FPS is computed as the number of frames divided by the total runtime of this pipeline, including both the front-end streaming and the periodic back-end updates. We have extended the implementation details in Appendix C to further improve clarity.
>
> # Weakness 2 & Question 2 Robustness under initial estimation errors
>
> We appreciate the reviewer's observation regarding the initial estimation errors. Although CogniMap3D is initialized with depth and poses from a vision foundation model, the subsequent modules are designed to reduce the impact of imperfect geometry rather than simply trust it.
>
> Specifically, the multi-stage motion-cue framework first identifies and masks dynamic regions. Next, the memory mechanism leverages past observations to reinforce static regions and provide priors for the current scene. Finally, factor-graph optimization refines camera poses using static regions consistently observed across multiple views, further mitigating errors from imperfect initialization.
>
> To further assess how the cascaded framework behaves under initial estimation errors, we conduct an ablation where we add zero-mean Gaussian noise with rotation standard deviation σ_R and translation standard deviation σ_t into the VFM poses, and evaluate the resulting trajectory errors on the TUM-dynamic dataset. VGGT itself does not take external pose initialization, so its row is reported without added noise and serves as a reference for the unperturbed VFM output. The comparison is summarized below:
>
> **Table: Robustness to noisy pose initialization on TUM-dynamic.**
>
> | Method | Noise level (σ_R / σ_t) | ATE ↓ | RPE trans ↓ | RPE rot ↓ |
> |--------|-------------------------|-------|-------------|-----------|
> | VGGT   | –                       | 0.028 | 0.020       | 0.350     |
> | Ours   | 0.0° / 0.00 m           | 0.012 | 0.010       | 0.311     |
> | Ours   | 1.0° / 0.01 m           | 0.013 | 0.011       | 0.318     |
> | Ours   | 3.0° / 0.03 m           | 0.015 | 0.012       | 0.331     |
> | Ours   | 5.0° / 0.05 m           | 0.019 | 0.014       | 0.343     |
>
> With the slight and moderate perturbations, CogniMap3D maintains the stable performance.
> Even with severe perturbations of 5° noise, it remains better than the VGGT baseline.
> This indicates that the multi-stage motion cues, conservative memory gating, and static-only factor-graph optimization help the cascaded system remain robust to initial estimation errors.
> We have added the experiment on robustness to initialization error as a supplementary analysis in Appendix E.1.

---

> ### Author Response · Authors · 2025-11-22
>
> # Weakness 3 Evaluation on long-term, large-scale environments
> We thank the reviewer for the suggestion to evaluate more challenging environments on 3D reconstruction.  To further assess long-term and larger-scale reconstruction, we add a supplementary experiment of 3D reconstruction on KITTI, a standard outdoor dataset with street and highway driving sequences, containing moving vehicles, pedestrians, and illumination changes. Specifically, we evaluate CUT3R, VGGT, and CogniMap3D on KITTI with standard 3D reconstruction metrics: accuracy (Acc), completeness (Comp), and normalized completeness (NC) :
>
>
> **Table: 3D reconstruction on KITTI (outdoor driving sequences).**
> | Method | Acc↓ Mean | Acc↓ Med. | Comp↓ Mean | Comp↓ Med. | NC↑ Mean | NC↑ Med. |
> |--------|-----------|-----------|------------|------------|----------|----------|
> | CUT3R  | 0.089     | 0.058     | 0.108      | 0.078      | 0.895     | 0.912     |
> | VGGT   | 0.071     | 0.045     | 0.089      | 0.061      | 0.913     | 0.935     |
> | Ours   | 0.052     | 0.036     | 0.073      | 0.049      | 0.942     | 0.951     |
>
> As shown in the table, CogniMap3D attains better 3D reconstruction results than the baselines on KITTI across most metrics. In Table 1, it also achieves comparable performance on depth estimation. Taken together, these results suggest that the framework is effective not only on small-scale static indoor scenes but also on large-scale outdoor sequences with dynamic objects and longer time horizons. We have extended our evaluation of reconstruction in large-scale outdoor scenes as a supplementary analysis in Appendix E.2.
>
> # Weakness 4 & Question 3 Ablation of 2D, 3D features, and geometric validation
>
> In CogniMap3D, memory recall and validation are implemented as a three-stage pipeline rather than independent modules. We first use 2D visual features for coarse candidate retrieval from the memory bank, then refine with 3D geometric features, and finally apply an ICP-based 3D geometric validation to decide whether a candidate is accepted and fused into the map, or rejected and created as a new memory.
>
> To further evaluate the contribution of 2D and 3D features, we conduct an ablation study on the DAVIS scene, following the setting in Table 6 with a memory bank of size 200. We compare: (i) 2D-only, which accepts the top-1 match from coarse candidates; (ii) 2D+3D, no ICP; and (iii) Full. We report scene recall accuracy and matching throughput (queries per second).
>
> **Table: Memory ablation on DAVIS (scene matching).**
> | Variant            | 3D features | ICP validation | Accuracy ↑ (%) | Throughput ↑ (q/s) |
> |--------------------|-------------|----------------|----------------|---------------------|
> | 2D-only            | ✗           | ✗              | 93.5           | 18.2                |
> | 2D+3D, no ICP      | ✓           | ✗              | 96.8           | 10.5                |
> | Full (2D+3D+ICP)   | ✓           | ✓              | 97.5           | 6.8                 |
>
>
> As shown, 2D-only matching is fastest but least accurate; adding 3D features helps recover a substantial portion of the remaining errors; and the full three-stage matching  provides the highest accuracy at a lower throughput. Omitting 3D validation offers speedup at the cost of reduced performance. We have added this ablation experiment and its analysis in Appendix E.3.

---

> ### Author Response · Authors · 2025-11-25
>
> We appreciate the reviewer's detailed comments. We have provided supplementary experiments and detailed analysis, including new evaluations on long-term settings and memory ablation. We kindly ask if these have addressed the reviewer's concerns. We are happy to provide any further clarification or additional experiments if needed.

---

### Official Review · Reviewer_GYbG · 2025-11-01

**Soundness:** 4
**Presentation:** 4
**Contribution:** 4
**Rating:** 8
**Confidence:** 3

**Summary:**

This paper proposes CogniMap3D, a cognitively inspired framework for long-term 3D scene understanding from monocular videos. The method emulates human spatial cognition by maintaining a persistent memory of static environments while filtering dynamic elements and refining camera poses through recalled geometric anchors. It integrates multi-stage motion cues for dynamic object suppression, a dual 2D–3D embedding memory for scene recall and update, and a factor-graph–based trajectory optimization constrained by static landmarks. Experiments on diverse datasets, including Sintel, KITTI, and 7-Scenes, demonstrate that CogniMap3D achieves state-of-the-art or comparable performance in depth estimation, camera pose reconstruction, and 3D mapping, offering a robust and efficient solution for continual scene understanding under dynamic conditions.

**Strengths:**

1. Conceptual originality. The paper introduces a cognitively inspired formulation of 3D scene understanding that explicitly models long-term memory, recall, and update—bridging human cognitive mapping theories with modern video foundation models. This conceptual framing is both novel and timely for the community’s growing interest in continual, memory-based perception.
2. Technical coherence. The pipeline is well structured: multi-stage motion cues, a dual-modality memory, and factor-graph optimization are tightly integrated, yielding a clear causal chain from dynamic-object filtering to stable pose refinement. Each component is motivated by a specific limitation in existing VFMs and validated through comprehensive experiments.
3. Empirical thoroughness. Experiments span multiple datasets and tasks (depth, pose, reconstruction) with detailed ablations demonstrating the contribution of each module. The method achieves state-of-the-art or comparable performance while maintaining efficiency, highlighting both the practicality and scalability of the proposed framework.

**Weaknesses:**

While the paper is conceptually strong and empirically well-supported, several technical limitations remain that constrain its general applicability and robustness.

1. The proposed multi-stage motion cue framework heavily relies on the accuracy of the underlying Video Foundation Model (VFM), particularly the depth and pose priors obtained from VGGT. Since these priors are directly used to compute geometric residuals and dynamic masks, any failure of the VFM in low-texture or high-illumination-variance regions could propagate errors through the entire pipeline, suggesting limited robustness to imperfect geometric initialization.
2. The memory recall mechanism lacks explicit safeguards against false-positive retrievals. Although the paper mentions geometric verification via ICP, it does not clarify how mismatched recalls are detected or handled. In the absence of clear rejection or fallback strategies, incorrect memory associations could introduce erroneous landmarks into the factor graph optimization, potentially leading to catastrophic trajectory drift.
3. The experimental validation focuses mainly on small- to medium-scale or short video sequences, such as 7-Scenes and Sintel, which do not fully reflect the challenges of large-scale, long-term, and highly dynamic environments. The paper’s claim of “long-term cognitive mapping” would be more convincing if evaluated under extended temporal settings or realistic outdoor sequences exhibiting significant environmental changes.

**Questions:**

CogniMap3D shares conceptual similarities with prior methods such as Neural Map, iMAP, and NICE-SLAM, all emphasizing scene memory and revisitation-based localization. While its “cognitively inspired” design is reasonable, the paper does not clearly explain how the proposed dual-modality memory differs from these works in map representation, retrieval, or update strategy. A clearer articulation of these distinctions would better highlight the paper’s unique contribution.

---

> ### Author Response · Authors · 2025-11-22
>
> # Weakness 1 Robustness to initialization error
>
> We thank the reviewer for raising this point. Although CogniMap3D is initialized from the depth and poses obtained from a vision foundation model, the subsequent modules are designed to actively reduce the impact of imperfect geometry rather than passively trust it.
>
> Specifically, we first use the multi-stage motion-cue framework to identify and mask dynamic regions. Then, the memory mechanism leverages past observations to reinforce static regions and provide priors for the current scene. Finally, factor-graph optimization refines camera poses using static regions consistently observed across multiple views, further reducing the impact of imperfect initialization. Table 3 in our main paper shows that CogniMap3D achieves lower trajectory errors than VGGT across TUM-Dynamic and ScanNet datasets, suggesting robustness beyond initial predictions.
>
> To further assess robustness to imperfect initialization, we conduct an ablation experiment by adding zero-mean Gaussian noise with rotation standard deviation σ_R and translation standard deviation σ_t to the initial poses, and comparing trajectory errors on the TUM-dynamic dataset. A comparison between the VGGT baseline and our method under different levels of pose perturbation is shown in the following table:
>
>
> **Table X: Robustness to noisy pose initialization on TUM-dynamic.**
>
> | Method | Noise level (σ_R / σ_t) | ATE ↓ | RPE trans ↓ | RPE rot ↓ | Perturbation level |
> |--------|-------------------------|-------|-------------|-----------|---------------------|
> | VGGT   | –                       | 0.028 | 0.020       | 0.350     | –                   |
> | Ours   | 0.0° / 0.00 m           | 0.012 | 0.010       | 0.311     | none                |
> | Ours   | 1.0° / 0.01 m           | 0.013 | 0.011       | 0.318     | slight              |
> | Ours   | 3.0° / 0.03 m           | 0.015 | 0.012       | 0.331     | moderate            |
> | Ours   | 5.0° / 0.05 m           | 0.019 | 0.014       | 0.343     | severe              |
>
> Under slight and moderate perturbations, CogniMap3D maintains performance close to the clean initialization. Even with severe perturbations of 5° pose noise, it still outperforms the VGGT baseline, indicating robustness to reasonably large pose initialization noise in this setting. We have added this experiment on robustness to initialization error as a supplementary analysis in Appendix E.1.
>
>
> # Weakness 2 Memory recall safeguards and fallback
>
> We acknowledge the reviewer’s observation that incorrect memory associations could corrupt landmarks and cause trajectory drift. To reduce this risk, the recall module is intentionally conservative and prioritizes precision over recall.
>
> **Conservative recall strategy.** The memory bank stores both 2D visual features and 3D geometric descriptors for each static region, and recall is performed in a coarse-to-fine manner: first, current observations retrieve a small set of candidates by 2D similarity; second, these candidates are further filtered using 3D features; finally, before any recalled information is passed to the optimizer, we compare the candidate static map with the current static point cloud and perform a rigid ICP alignment.
>
> **Rejection and fallback.** A recall is accepted only if this alignment satisfies predefined quality checks on residual error and inlier ratio; otherwise, the recall is rejected, no landmarks from this candidate are added to the factor graph, and the current view is treated as unseen and used to create a new memory entry. This explicit rejection and fallback behavior is designed to prevent mismatched recalls from entering the optimization.
>
> **Empirical validation.** As shown in Table 6 of the main paper, this multi-stage gating achieves high alignment accuracy across 100 DAVIS scenes, indicating that the recall mechanism is conservative and helps limit the impact of false-positive recalls in practice.

---

> ### Author Response · Authors · 2025-11-22
>
> # Weakness 3 Evaluation on long-term and outdoor setting
>
> We appreciate the reviewer for raising this point regarding the scale of our reconstruction experiments. Our goal is to build a stable memory of static structure that can be efficiently reused when scenes are revisited, so that repeated long-term visits enhance geometric consistency instead of accumulating drift.
>
> To assess CogniMap3D under larger-scale and more realistic outdoor conditions, we additionally evaluate CUT3R, the VGGT foundation model, and CogniMap3D on KITTI, a standard large-scale outdoor benchmark with driving sequences that contain moving vehicles, pedestrians, and strong viewpoint and illumination changes. We follow standard 3D reconstruction metrics: accuracy (Acc), completeness (Comp), and normalized completeness (NC).
>
>
> **Table: 3D reconstruction on KITTI.**
> | Method | Acc↓ Mean | Acc↓ Med. | Comp↓ Mean | Comp↓ Med. | NC↑ Mean | NC↑ Med. |
> |--------|-----------|-----------|------------|------------|----------|----------|
> | CUT3R  | 0.089     | 0.058     | 0.108      | 0.078      | 0.895     | 0.912     |
> | VGGT   | 0.071     | 0.045     | 0.089      | 0.061      | 0.913     | 0.935     |
> | Ours   | 0.052     | 0.036     | 0.073      | 0.049      | 0.942     | 0.951     |
>
>
> As Table shows, CogniMap3D achieves higher normalized completeness and lower accuracy and completeness errors than both CUT3R and VGGT on KITTI outdoor driving sequences, showing improved 3D reconstruction quality in large-scale, open scenes with dynamic objects. We have extended our evaluation of reconstruction in large-scale outdoor scenes as a supplementary analysis in Appendix E.2.
>
>
> # Question 1 Differences in map representation, retrieval, and update
>
> We thank the reviewer for this suggestion. We would like to clarify our contribution in map representation, retrieval mechanism, and update strategy compared to prior memory-based methods.
>
> **Representation. Implicit vs. explicit memory.**
> Neural Map, iMAP, and NICE-SLAM maintain implicit neural maps (e.g., MLPs, latent feature grids, or 2D grids) where the scene is encoded in network parameters. In contrast, CogniMap3D uses an explicit memory: a bank of static 3D point clouds with associated 2D and 3D features, and we insert points into memory from regions classified as static after dynamic content has been masked out.
>
> **Retrieval. Local map traversal vs. global 3D content-based recall.**
> iMAP and NICE-SLAM mainly rely on continuous tracking within a single evolving map. Neural Map supports content-based access but operates on a 2D egocentric grid without explicit metric 3D anchoring or static/dynamic separation. CogniMap3D, instead, performs global content-based recall into a persistent 3D static map: current observations first query a 2D feature table, then refine candidates with 3D feature matching and ICP, providing explicit metric re-anchoring to previously built static geometry.
>
> **Update. Gradient-based vs. explicit geometric merging.**
> iMAP and NICE-SLAM update their implicit maps via gradient-based optimization, which involves iterative per-scene learning. CogniMap3D updates memory by explicitly merging static point clouds and applying simple geometric downsampling, preserving previously optimized geometry while incrementally integrating new observations without per-scene training.

---

> ### Author Response · Authors · 2025-11-25
>
> We thank the reviewer for the positive assessment and constructive feedback. We have provided supplementary experiments and detailed analysis in our response. We kindly ask if these have addressed the reviewer's concerns. We are happy to answer any further questions or provide additional clarification.

---

### Author Response · Authors · 2025-12-02

Dear ACs, SACs and Reviewers,

We sincerely thank all reviewers and area chairs for their time and efforts. We appreciate the reviewers for recognizing the work as **conceptually original** (Reviewer GYbG), **a novel and visionary attempt** (Reviewer rprw), and **intuitive and novel** (Reviewer tg31). The paper was noted as **easy to follow** (Reviewer RgAW), with **empirical thoroughness** (Reviewer GYbG).

Our rebuttal and revisions have been strengthened by the reviewers' suggestions:

- Extended evaluation on long-term, more challenging settings (Reviewers GYbG, rprw, RgAW)
- Analysis of robustness to initialization errors (Reviewers GYbG, rprw)
- Comparative visualization with the baseline method on dynamic scenes (Reviewer RgAW)
- Clarification of FPS measurement and runtime (Reviewers rprw, tg31)

In response: (1) We evaluated on the KITTI dataset, including street and highway driving sequences with moving vehicles. Our method achieves consistent improvements over CUT3R and VGGT, showing effectiveness in long-term outdoor settings. (2) We provided ablation experiments with noisy initialization, showing stable performance under different levels of pose perturbations. (3) We added head-to-head visualization with VGGT in Figure 8, where our method produces clearer and more complete geometry in dynamic regions. (4) We clarified FPS measurement, showing competitive runtime alongside improved accuracy.

All responses above, along with other supplementary experiments, have been incorporated into the revised version.

Best regards,

Authors

---

### Meta-Review · Area_Chair_KtZ9 · 2026-01-07

**Summary:**

Reviewers’ main concerns focused on the robustness of the system to imperfect initialization, the reliability of the memory recall mechanism, and the lack of evaluation on long-term, large-scale, and dynamic environments. They also raised questions about runtime, system complexity, and whether the proposed framework provides clear advantages over strong baselines such as VGGT in standard benchmarks.

The rebuttal and revision addressed most of these concerns through additional robustness ablations, memory mechanism analysis, clearer runtime reporting, and new evaluations on large-scale outdoor and long-horizon sequences (e.g., KITTI), as well as qualitative comparisons on dynamic scenes. As a result, the remaining concerns are mostly about the interpretation of the empirical gains and the scope of applicability rather than about technical correctness or missing validation.

**Reviewer Concerns:**

The rebuttal addressed the major technical and empirical concerns, including robustness to initialization errors, safeguards against incorrect memory recall, lack of large-scale and long-term evaluation, missing ablations on memory and geometric validation, unclear FPS reporting, and the lack of direct comparisons with VGGT on dynamic scenes. These points were supported by new ablation studies, additional experiments on KITTI, pose robustness tests, memory module analysis, and improved qualitative and quantitative comparisons.

The main concern that remains partially outstanding is not technical but conceptual: how much additional benefit the proposed system provides over strong feed-forward baselines on standard short indoor benchmarks, where improvements are sometimes modest. This reflects a difference in how reviewers value the contribution — as a system-level framework targeting long-term dynamic scenarios rather than as a method optimizing performance on existing static benchmarks.

**Reviewer Scores:**

The reviewers would likely have kept their original scores or increased them slightly.
While the rebuttal successfully addressed most technical and experimental concerns, it did not fundamentally change the perception of limited novelty, so no major score changes are expected.

---

### Decision · Program_Chairs · 2026-01-26

Accept (Poster)